# Stream Data Load Prediction for Resource Scaling Using Online Support Vector Regression

**Zhigang Hu, Hui Kang** 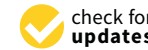 **and Meiguang Zheng** *

School of Software, Central South University, Changsha 410075, China; zghu@csu.edu.cn (Z.H.); huikang@csu.edu.cn (H.K.)

**\*** Correspondence: zhengmeiguang@csu.edu.cn; Tel.: +86-137-55101439

**Abstract:** A distributed data stream processing system handles real-time, changeable and sudden streaming data load. Its elastic resource allocation has become a fundamental and challenging problem with a fixed strategy that will result in waste of resources or a reduction in QoS (quality of service). Spark Streaming as an emerging system has been developed to process real time stream data analytics by using micro-batch approach. In this paper, first, we propose an improved SVR (support vector regression) based stream data load prediction scheme. Then, we design a spark-based maximum sustainable throughput of time window (MSTW) performance model to find the optimized number of virtual machines. Finally, we present a resource scaling algorithm TWRES (time window resource elasticity scaling algorithm) with MSTW constraint and streaming data load prediction. The evaluation results show that TWRES could improve resource utilization and mitigate SLA (service level agreement) violation.

**Keywords:** streaming processing; dynamic prediction; auto-scaling; online support vector regression; time window maximum throughput

## 1. Introduction

With the development of the Internet, Internet of things and big data technology, streaming data, which is the infinite, real-time and dynamic stream of data, has been applied more and more for the fields of financial analysis, social media, sensors, websites, large-scale scientific experiments and so on. Such data is quickly analyzed to get the most value, and traditional data management systems cannot process such large amounts of streaming data in real time. Aiming to achieve the characteristics of infinite, real-time, orderly and large-scale data flow, the industry has designed and developed storm, Spark and other streaming processing systems to process real-time mutation data-flow in an efficient and scalable manner.

Cloud computing can add on-demand elasticity to these stream processing systems to deal with fluctuating computing demand using autoscaling [1]. To guarantee a service level agreement (SLA) in terms of application performance, we need a prediction model that connects VM configurations and application performance so that the auto-scalers can estimate and provision the right number of VMs. Imai. et al. [2] proposed a maximum sustainable throughput (MST) prediction models for stream processing systems. The models assume a homogeneous VM instance type and predict MST only by the number of VMs, but they do not consider the time window interval performance. The paper strengthens the MST to the maximum sustainable throughput of time window (MSTW). If the incoming data rate exceeds the system's MSTW, unprocessed data accumulates, eventually making the system inoperable. By dynamically allocating and deallocating VMs, service providers can make sure the MSTW remains larger than the incoming data load, to maintain stable service operation. Techniques that have been used to predict data load time series include autoregressive moving average

model ARMA [3], neural network [4], and support vector regression SVR [5]. However, these methods are not real online predictions and cannot handle real-time stream data. Our solution responds to the problem by applying a more accurate online SVR algorithm to predict the incoming data-flow load and then measuring the number of virtual machines for the maximum sustainable throughput of time window (MSTW). The main contributions are as follows:

- Improving online SVR prediction algorithm to predict load time series of stream data to reduce delay of cloud resource distribution.
- According to the maximum throughput calculated by MSTW, and the threshold rule is set up with the prediction value of the stream data load to put forward an elastic resource distribution algorithm to guarantee the system's SLA.

## 2. Related Work

At present, research on streaming big data has continuously expanded and deepened on the international plane, and research on real-time prediction of stream data load and elastic resource allocation has attracted people's attention.

For instance, (Li at al [6]; Tiziano et al. [7]; Lohmann et al. [8]) proposed some resource distribution strategies related to "delay detection", but they were not considering any other metric. (Fernandez et al. 2013) [9] showed that trigger elastic adjustments based on CPU utilization, memory, or network resource setting threshold rules. Unlike the timing of the "delay detection" strategy, the "pre-adjustment" strategy is based on load forecasting to adjust resources ahead of time and can improve the efficiency of resource distribution. Liu et al. [10] proposed a cloud resource allocation framework, where the global layer uses deep reinforcement learning to deal with complex control problems, and local layer uses LSTM to predict workloads. This achieved the best tradeoff between delay, power consumption and energy consumption in a cloud server cluster. Fu et al. [11] proposed a dynamic resource scheduling system, which includes the he performance model of queuing theory, and can dynamically configure and schedule the real-time response of cloud resources to achieve the best resource consumption. Mayer et al. [12] used time series analysis to predict the input data of the operation and calculates the parallelism of operation based on queuing theory. Kumbhare et al. [13] considered the volatility of resource performance and used a heuristic resource adjustment method to maintain the throughput of the application at the minimum resource cost. Kumbhare et al. [14] proposed a method to predict the stream data load and performance, which adaptively plans resource distribution to limit throughput while satisfying resource adjustment costs. Ye et al. [15] attempted to anticipate the micro-architecture-level interference by using an offline profiling phase. Rao et al. [16] proposed an effective metric to predict the performance of applications running in a NUMA system. Such a metric can be leveraged to design a resource allocation that is aware of contention among shared resources. However, obtaining such an interference signature through profiling might not be feasible in every practical case, as the interference attributes of applications could change over the run-time. Madsen et al. [17] integrated fault-tolerance and dynamic resource managements to avoid waste from excessive processing delay. Wu et al. [18] introduced Chrono Stream, which provides vertical and horizontal elasticity. Mencagli [19] proposed a multi-input, multi-output (MIMO) resource controller that automatically adapts to dynamic changes in a shared infrastructure. Such models try to estimate the complex relationship between the application performance and the resource allocation and adjust the embedded model by measuring the clients' response time. Ruan et al. [20] presented an approach based on monetary cost model for cloud resource provisioning for Spark with sample running, which inspires us to leverage the historical information of previous batches in batched streaming system to make resource allocation decision in our design. Park et al. [21] dynamically increased or decreased the computing capability of each node to enhance locality-aware task scheduling with dynamic virtual CPU number configurations.

Recently, some new algorithms for solving different problems of various fields have appeared in the artificial intelligence and reputable Operations Research field. Canales-Bustos et al. [22]

suggested a Pareto-based algorithm, which is a Multi-objective Hybrid Particle Swarm Optimization metaheuristic for the design of an effective decarbonized supply chain in mining. A novel heuristic algorithm, which accounts for the truck service priority and the truck service order restrictions, was proposed for initializing the chromosomes and population [23]. Peres et al. [24] presented an exact formulation and developed a metaheuristic to solve a multi-period, multi-product Inventory-Routing Problem. A Memetic Algorithm was developed to minimize the total vessel service cost for the vessel diversion decision making [25]. Dulebenets et al. [26] proposed a hybrid method based on a Lagrangian relaxation technique through the volume algorithm. Using information from the Lagrangian multipliers, constructive heuristics with local search procedures were utilized to reach the objective of minimizing the makespan in cross-docking. Fonseca et al. [27] developed a metaheuristic hybrid based on adaptive large neighborhood search and tabu search, called ALNS/TS, which assume a higher number of customer orders and higher capacities of the picking device. Žulj et al. [28] proposed a multi-objective mixed integer nonlinear optimization model for the vessel scheduling problem.

A few Evolutionary Algorithms had been developed to solve the stream processing and resource allocation problem. Aiming at the characteristics of real-time streaming data and higher value of new data, this paper proposes a new online support vector regression algorithm, which gives different values to different time periods and improves the accuracy of data load prediction.

## 3. Task Description

### 3.1. System Aarchitecture

The system architecture of this paper is shown in Figure 1. The Producer sends data to Kafka, which is an efficient distributed messaging system in period $t$. Spark extracts data from Kafka in real time and handles it at the interval of time window $t$. A Spark cluster contains multiple work nodes, each of which contains one or more executor. Producer sends data to Kafka in real time, and Kafka stores the data in Partition as a unit. After Spark draws the Kafka data, Spark converts this data into a series of RDD for processing, and the internal structure of RDD is also presented in a partitioned form. Processing of RDD partition data within a time window $t$ should be submitted to executor in the Spark cluster work node and execute it as a task. An RDD data corresponds to the cumulative data that Spark reads from Kafka in a time slice. The number of partitions in Kafka, partition in RDD and the number of tasks executed in the end must be consistent. There are three core processes of the system framework: the load forecasting model, the performance model, the flexible resource distribution. The load forecasting model predicts the load trend of the next window $t$ according to historical data of the stream data load. The performance model is obtained by monitoring the throughput of the Spark processing data and the batch response time and so on. The flexible resource distribution will get the timing of the flexible operation based on the relation of load of next $t$ and the performance model, and assigns a reasonable number of virtual machines to the Spark cluster before the next time window $t$.

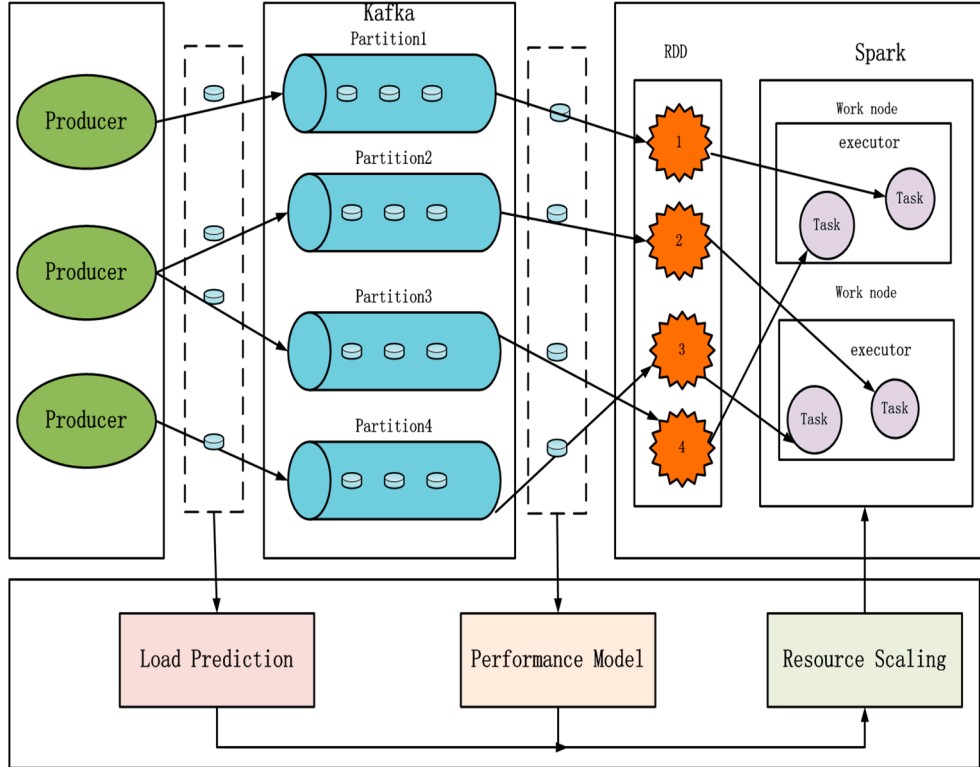

**Figure 1.** The article system architecture.

### 3.2. Time Series Prediction of Stream Data Load

As the data source, the Producer sends data to Kafka in real time in time window *t*, and the stream data load is a continuously changing data sequence in time period *t*. Stream data load is modeled as time series $X = \{x_1, x_2, x_3, \ldots, x_n\}$ in this paper. The first core of Figure 1 predicts the stream data load time series and can reduce the potential SLA violation and the delay of resource distribution. The time series prediction is based on the sequence *X* of an observation value of the past and finds a function *f* (*) that conforms to the law of the change of the system. According to this function, the past values are used as input to predict the future value and output. Given the time series of stream data load as *X*, the prediction of stream data load in next time window *t* can be described as $x_{n+t} = f(x_n, x_{n-1}, \ldots, x_{n-m+1})$.

The Online SVR method can learn and update the data online. It is suitable for solving the problem of complex data characteristics and forecasting of dynamic updating of the model. In this paper, the improved online SVR algorithm is used to predict the input load of the stream data in the next period. That is, an improved online SVR is used to fit *f* (*) and predict the future value.

## 4. Improved Online SVR for Stream Data Load Time Series Prediction

The stream data attributes of network log include time record, IP address, username, request type, state, load (with wordcup98 as an example). In this paper, the time record and load are selected as the research attributes. The historical stream dataset is represented as $Z = \{(x_1, y_1), (x_2, y_2), \ldots, (x_i, y_i)\} \in (X \times Y)^l$. **X** represents the time feature vector, and **Y** represents the stream data load feature vector. When stream data in the past is set as training samples, constructing an online SVR prediction function as $f(x) = w^T \phi(x) + b$. The nonlinear mapping $\phi(*)$ represents the feature vector after the mapping of the *x*. It maps the feature space of the input stream data set to high dimensional feature space, so that the function in the high dimensional feature space can be expressed as a linear regression function. *w* represents the normal vector of hyperplane in time eigenvector space, and it determines the direction of hyperplane. *b* is the displacement term, and it determines the distance between the hyperplane

and the origin. Obviously, the partition hyperplane is determined by the normal vector $w$ and the displacement $b$. If you want to get an accurate predictive function, you need to find the maximum interval distance of hyperplane. Therefore, we write the problem into a convex optimization problem:

$$
\min \frac{1}{2}\|w\|^2
$$
$$
s.t. \begin{cases} y_i - w^T \phi(x_i) - b \le \varepsilon \\ w^T \phi(x_i) + b - y_i \le \varepsilon \end{cases}, i = 1, \dots, l \tag{1}
$$

when solving $w$ and $b$, the standard online SVR algorithm considers the same scale for error items, this results in poor prediction of stream data items with different variance of error items $\delta_i^2$.

To solve this problem, the paper introduces an appropriate weight $\lambda_i$ to adjust the role of each error item in regression. In addition, the standard online SVR uses the same insensitive loss $\varepsilon$ for all data points. However, the timeliness of streaming data is very strong, the new data is more instructive than the old data, different samples should use different $\varepsilon$. In this paper, $\varepsilon$ of each data uses different weights $(1-d)^i$, the relaxation factor $\tau_i \ge 0$ and $\tau_i^* \ge 0$ as well as the weight $\lambda_i = 1/\delta_i^2$ and $p^i$ are introduced to solve $w$ and $b$:

$$
\min \frac{1}{2}\|w\|^2 + C \sum_{i=1}^{l} \lambda_i (\tau_i + \tau_i^*)
$$
$$
s.t. \begin{cases} y_i - w^T \phi(x_i) - b \le p^i \varepsilon + \tau_i \\ w^T \phi(x_i) + b - y_i \le p^i \varepsilon + \tau_i^* \end{cases}, i = 1, \dots, l \tag{2}
$$

where $C$ is the penalty parameter, $\varepsilon$ is the insensitive loss function, $d$ is the change factor between 0 and 1. the weights $p^1$ correspond to the data furthest from the prediction point, and $p^i$ correspond to the data closest to the prediction point.

The Lagrange multiplier method is used to solve the formula (2). First, the constraint conditions are fused into the target function by the Lagrange function, and the Lagrange equation is obtained.

$$
L(w, b, \alpha, \alpha^*, \beta, \beta^*) = \frac{1}{2}\|w\|^2 + C \sum_{i=0}^{l} \lambda_i(\tau_i + \tau_i^*) - \sum_{i=1}^{l}(\beta_i \tau_i + \beta_i^* \tau_i^*) + \sum_{i=1}^{l} \alpha_i(w^T \phi(x_i) - p^i \varepsilon - y_i - \tau_i)
$$
$$
+ \sum_{i=1}^{l} \alpha_i^*(y_i - w^T \phi(x_i) - p^i \varepsilon - \tau_i^*) \tag{3}
$$

Here, $(\alpha, \alpha^*, \beta, \beta^*) \ge 0$ is the Lagrange multiplier. The Lagrange equation makes partial derivatives for $w$, $b$, while $\tau_i$ and $\tau_i^*$ make it zero:

$$
\frac{\partial L}{\partial W} = w - \sum_{i=1}^{n}(\alpha_i^* - \alpha_i)x_i = 0 \tag{4}
$$

$$
\frac{\partial L}{\partial b} = \sum_{i=1}^{n}(\alpha_i^* - \alpha_i) = 0 \tag{5}
$$

$$
\frac{\partial L}{\partial \tau_i} = C\lambda_i - \beta_i - \alpha_i = 0 \tag{6}
$$

$$
\frac{\partial L}{\partial \tau_i^*} = C\lambda_i - \beta_i^* - \alpha_i^* = 0 \tag{7}
$$

Considering feature mapping and introducing kernel functions, formula (4) will be shaped as

$$
w = \sum_{i=1}^{l}(\alpha_i^* - \alpha_i)K(x_i, x_j) \tag{8}
$$

Here, kernel function $K(*) = \phi(*)\phi(*)^T$ is introduced, which makes the function solution bypass the feature space and obtain the function directly in the input space, and avoid the simplified calculation of the nonlinear mapping $\phi(*)$.

Substituting (4)–(7) into (3) to obtain the duality problem of the problem:

$$\max \sum_{i=1}^{l} y_i(\alpha_i^* - \alpha_i) - p^i \varepsilon(\alpha_i^* + \alpha_i) - \frac{1}{2} \sum_{i,j=1}^{l} (\alpha_i - \alpha_i^*)(\alpha_j - \alpha_j^*) K(x_i, x_j)$$

$$s.t. \begin{cases} 0 \le \alpha_i, \alpha_i^* \le \lambda_i C \\ \sum_{i=1}^{l} (\alpha_i - \alpha_i^*) = 0 \end{cases} \tag{9}$$

The conversion of dual problems needs to meet KKT conditions:

$$\begin{cases} \alpha_i(w^T \phi(x_i) - p^i \varepsilon - y_i - \tau_i) = 0 \\ \alpha_i^*(y_i - w^T \phi(x_i) - p^i \varepsilon - \tau_i^*) = 0 \\ \alpha_i \alpha_i^* = 0, \tau_i \tau_i^* = 0 \\ (\lambda_i C - \alpha_i)\tau_i = 0, (\lambda_i C - \alpha_i^*)\tau_i^* = 0 \end{cases} \tag{10}$$

According to the KKT conditions, when $0 \le \alpha_i, \alpha_i^* \le \lambda_i C, \tau_i = 0$, then

$$b = y_i + p^i \varepsilon - \sum_{i=1}^{l} (\alpha_i^* - \alpha_i) K(x_i, x_j) \tag{11}$$

Improving the initialization model parameters of online SVR is the key point to complete the follow-up prediction work. The improved online SVR model is initialized after the solution is completed. Through model training, the boundary support vector function of sample $x_i$ at the time point $i$ is defined as:

$$h(x_i) = f(x)^* - y_i \tag{12}$$

We set $\theta_i = \alpha_i - \alpha_i^*$, based on Equation (5) to (8):

$$\begin{cases} h(x_i) \ge p^i \varepsilon, \theta_i = -\lambda_i C \\ h(x_i) = p^i \varepsilon, \theta_i \in [-\lambda_i C, 0] \\ h(x_i) \in [-p^i \varepsilon, p^i \varepsilon], \theta_i = 0 \\ h(x_i) = -p^i \varepsilon, \theta_i \in [0, \lambda_i C] \\ h(x_i) \le -p^i \varepsilon, \theta_i = \lambda_i C \end{cases} \tag{13}$$

According to the formula (13), the stream data training set can be divided into three subsets, namely, the error support vector set E, the support vector set $S$ and the preserved sample set $R$:

$$\begin{aligned} S &= \left\{ i \middle| (\theta_i \in [0, \lambda_i C] \wedge h(x_i) = -p^i \varepsilon) \vee (\theta_i \in [-\lambda_i C, 0] \wedge h(x_i) = p^i \varepsilon) \right\} \\ E &= \left\{ i \middle| (\theta_i = \lambda_i C \wedge h(x_i) \ge p^i \varepsilon) \vee (\theta_i = \lambda_i C \wedge h(x_i) \le p^i \varepsilon) \right\} \\ R &= \left\{ i \middle| (\theta_i = 0 \wedge h(x_i) \le p^i \varepsilon) \right\} \end{aligned} \tag{14}$$

When the stream data is updated online, the new stream data payload value $x_c$ is added to the training set. Now $\theta_c$ is added, and the $\theta_i$, $\theta_c$ of the three subsets of the training set are updated directly so as to get the updated values $\Delta \theta_i$ and $\Delta \theta_c$, then it is still satisfying (13) and an online SVR regression model with an online dynamic update is obtained.

$$f(x) = \sum_{i=1}^{m} (\alpha_i - \alpha_i^*) K(x_i, x_j) + b \tag{15}$$

## 5. Stream Data Load Prediction for Cloud Recourse Scaling

For the data load that changes with time, fixed cloud resource distribution will result in a waste of resources or a low quality of service. Based on stream data load time series prediction, this paper proposes a flexible distribution algorithm to extend or shrink the virtual machine in order to satisfy SLA. The maximum sustainable throughput (MST) is the maximum throughput of the stream processing system that can read the data indefinitely. It is an important indicator that the cloud service provider needs to consider in terms of stream processing system. The MST performance model is a function of system MST and virtual machine. It is defined as follows:

$$MST(m) = \frac{1}{t(m)} = \frac{1}{w_0 + w_1 * \frac{1}{m} + w_2 * m + w_3 * m^2} \tag{16}$$

Here, $m$ represents the number of virtual machines, and $w_0$ represents serial processing time, $w_1$ represents parallel processing time, $w_2$ represents data input / output time, $w_3$ indicates the communication time between virtual machines, and all weights are non-negative ($w_i \geq 0, i = 0, 1, 2, 3$).

While the stream processing system resources are fixed, the number of work nodes and the number of executor assigned in the work node are fixed, and the number of task that can be processed is fixed, when the stream data input load is larger than throughput of the system, it will lead to the accumulation of large amounts of data in Kafka, and the stream processing system cannot process data normally. The spark processing data are handled at the interval of the time window. When the time window is not considered while configuring the number virtual machines appropriately by using the MST performance model, only the overall throughput of the system is greater than the input load. This may result in a system throughput of less than input load at the interval of the time window, the next time window system throughput is far greater than the input load. From the overall point of view, the system processing data is greater than the input load, which will lead to the system in a certain time window not being able to handle the normal data. However, if the data is lost to the next time window to process, it may cause a data processing delay, disorderly sequences, packet loss and other consequences. In order to ensure the performance of the system, in each time window interval, the system throughput is greater than its input load, so the overall throughput of the system will be larger than the overall input load, and the stream processing system can handle the data normally. Therefore, it is useful and necessary to get the maximum throughput of data in the time window interval. The paper strengthens the MST to the maximum sustainable throughput of time window (MSTW), MSTW is the limit of the flow processing system's sustainable processing of data within the time window, and the MSTW performance model is calculated as follows:

$$MSTW(m, T) = MST(m) * T \tag{17}$$

Here, $T$ refers to the interval of the time window.

The MSTW performance model obtains the maximum throughput of the virtual machine within the time window interval, which can handle the maximum throughput of $MSTW_{SLA}(m, T)$. To expand the cluster to deal with the fluctuating stream data load, we can predict the data input load in advance by understanding the relationship between VMs and the MSTW. The cluster can distribute the VMs flexibly to stabilize the processing performance of the system before the stream processing system reaches MSTW. This paper presents an elastic allocation algorithm TWRES (Time window resource elasticity scaling algorithm) (Algorithm 1) for load forecasting and time window throughput.

With the application $A$, the time window interval $T$, the response time limit $R_{SLA}$, the application average response time $R(A, T)$, the payload prediction value $F$ together as the input of the flexible distribution algorithm; the output is the number of extended virtual machine $UbestVMs$ or the contractile virtual machine $DbestVMs$.

The algorithm first initializes the application average response time, and then compares it with the load prediction value $F$ obtained by the load forecasting algorithm. If the $F$ value is larger, then the

existing cluster resources are unable to handle the overloaded stream data. We need to add virtual machine source processing data to the cluster. If the *F* value is smaller, it shows that the system can handle the data normally, but the data is delayed in the process of waiting for processing and we can calculate the average time of delay using $R(A, T)$. If it is larger than the set $R_{SLA}$, some data cannot be processed for a long time, and we assign the value one. The platform virtual machine handles these data and reduces the delay in application execution. On the contrary, if the latency is not exceeded, the virtual machine is reclaimed to reduce the waste of resources, and min*VMs* is the minimum number of virtual machines that can ensure that the $MSTW_{SLA}(m, T)$ is greater than *F*.

---

**Algorithm 1.** Time Window Resource Elasticity Scaling Algorithm

---

**Input**: A, T, $R_{SLA}$, $R_{(A,T)}$, F, $MSTW_{SLA}(m, T)$
**Output:** *UbestVMs* or *DbestVMs*
1) Begin
2) $r = R(A, T)$
3) if $F \geq MSTW_{SLA}(m, T)$
4) 　$UbestVMs \leftarrow \min VMs - m$
5) else if $F < MSTW_{SLA}(m, T)$
6) 　　if $r > R_{SLA}$
7) 　　　$UbestVMs \leftarrow 1$
8) 　　else
9) 　　　$DbestVMs \leftarrow m - \min VMs$
10) End

---

## 6. Results

### 6.1. Experimental Setup

In order to study the accuracy of the algorithm and the flexibility of the flexible distribution strategy, the experiment runs on 10 nodes. Each node is configured as follows: 2 nuclear CPU, 8 GB memory, and Centos7.0 X86_64 control system. The Apache Spark stream processing system is selected in the experiment because of its flexible deployment on the available machine. Hadoop3.0, Spark2.3, Scala2.10, Kafka1.0 and JDK/JREv1.8 build Spark cluster are also installed in each node.

### 6.2. Estimation of Stream Data Load Prediction Algorithm

In order to verify the effectiveness of online SVR for the prediction algorithm of stream data input load time series, 4 open, nonlinear and non-stationary time series data sets, namely WordCup98 [29], Poland Electric Demand time series, Darwin Sea Level Pressures, and Darwin Sea are tested [30]. Four time series datasets are used to carry out 160 step single step prediction, and the prediction efficiency of online SVR algorithm, the improved online SVR algorithm and BPNN algorithm is compared comprehensively.

Parameter selection generally adopts the method of combining cross-verification with grid search. The existing method of combining cross-validation with grid search predicts each set of parameters together on each subset and takes the parameter that ensures its average error is as small as possible. However, the object predicted in this paper is the stream data time series, which uses historical data to predict future data. Therefore, using the follow-up data to predict the previous data dose not consist characteristics of stream data series in cross-validation. This paper improves the cross-validation method. By using a subset of $1 - n$ to test subset of $n + 1$, it can improve the accuracy and stability. The specific experimental conditions are as follows: The Gauss kernel function is selected in an improved Online SVR algorithm, the penalty parameter $C_G = 10$, the insensitive loss function kernel function $\varepsilon_G = 0.1$. Parameter $H_G = 20$, and the length of the initial online modeling time sequence is 100. The online SVR algorithm selects the RBF kernel function, with the penalty parameter $C_R = 1$, the insensitive

loss function kernel function $\varepsilon_R = 0.1$. Parameter $H_R = 20$, and the initial modeling time series length is 100. BPNN selects 3 layers and using sliding window method to train the network. The activation function of the forward process selects the Tansig function and the Purelin function. Target accuracy is 0.005, maximum cycle number is 5000.

In the experiment, two average standards are adopted to make comprehensive comparison between algorithms: Mean Absolute Error, (MAE) and Normalized Root Mean Square Error, (NRMSE). The definitions of MAE and NRMSE are as follows:

$$MAE = \frac{1}{n}\sum_{i=1}^{n}\left|x(i) - x(i)^*\right| \tag{18}$$

$$NRMSE = \frac{\sqrt{\frac{1}{n}\sum_{i=1}^{n}\left[x(i) - x(i)^*\right]^2}}{\sqrt{\frac{1}{n}\sum_{i=1}^{n}\left[x(i) - x^*\right]^2}} \tag{19}$$

$n$ is the number of data points of the dataset, $x(i)$ is the true value, $x(i)^*$ is the predicted value, and $x^*$ is the mean of the time series.

The results of four time series prediction accuracy and execution efficiency are shown in Table 1. The improved online SVR algorithm is superior to the standard online SVR algorithm and BPNN in prediction accuracy, the execution efficiency is higher, and the execution efficiency is improved by about 20%–30%. Therefore, the improved online SVR algorithm can predict the time series very well.

**Table 1.** Time series forecast results.

| Series | Data Set | Algorithm | MAE | NRME | Time |
|--------|----------|-----------|-----|------|------|
| 1 | WordCup98 | online SVR | 0.082 | 0.455 | 0.135 |
| | | Improved online SVR | 0.001 | 0.106 | 0.124 |
| 2 | Poland Electric | BPNN | 0.054 | 0.325 | 0.130 |
| | | online SVR | 0.112 | 0.452 | 0.263 |
| | | improved online SVR | 0.049 | 0.311 | 0.116 |
| 3 | Darwin Sea | BPNN | 0.1336 | 0.006 | 0.200 |
| | | online SVR | 0.134 | 0.456 | 0.345 |
| | | improved online SVR | 0.0551 | 0.366 | 0.321 |
| 4 | Sunspot database | BPNN | 0.123 | 0.426 | 0.369 |
| | | online SVR | 0.162 | 0.564 | 0.334 |
| | | improved online SVR | 0.124 | 0.521 | 0.316 |
| | | BPNN | 0.154 | 0.86 | 0.468 |

*6.3. Estimation of Stream Data Load Prediction Algorithm*

In order to verify the accuracy of the MSTW model, we first need to train the parameters of the model. We run the four benchmarks, namely Grep, Rolling Count [31], Unique Visitor, Page View, and Rolling Sort on the cloud platform and collect the MSTW when M = {1,2,3,4,5,6,7,8,9,10}, and set the number of virtual machine used to verify the model as Test of M = {7,8,9,10}. For Grep and Rolling Count, The Adventures of Tom Sawyer document data [32] is used as input, as well as Unique Visitor and random analog access. Input data payload and data processing throughput can be obtained through Java Management Extensions (JMX) monitoring. The time window T is set as 5 seconds. The parameters obtained from model learning are shown in Table 2, and $w_2$, $w_3$ are always 0. This shows that the model we have developed is a linear model.

**Table 2.** Model learning parameters.

| Benchmark | $w_0$ | $w_1$ | $w_2$ | $w_3$ |
|---|---|---|---|---|
| Grep | 0.0272 | 0.2172 | 0 | 0 |
| Rolling Count | 0.04933 | 0.33 | 0 | 0 |
| Unique Visitor | 0.009 | 0.312 | 0 | 0 |
| Page View | 0.004 | 0.576 | 0 | 0 |

After learning the model parameters, the model can be used to predict the MSTW of a greater number of virtual machines, and the MSTW prediction results of different number of virtual machines are shown in Figure 2. As can be seen from the diagram, the prediction accuracy of the MSTW model of Unique Visitor is better. The prediction of Page View has a larger deviation when m = 10, and the prediction accuracy is higher for the number of other virtual machines. However, when the model is predicted in Grep, the overall prediction value is lower than the actual value, but it still has a better fit for the trend of throughput. Though the effect is not as good as the other three in the prediction of the pair and Rolling Count, the prediction value is also close to the real value when m = 8.

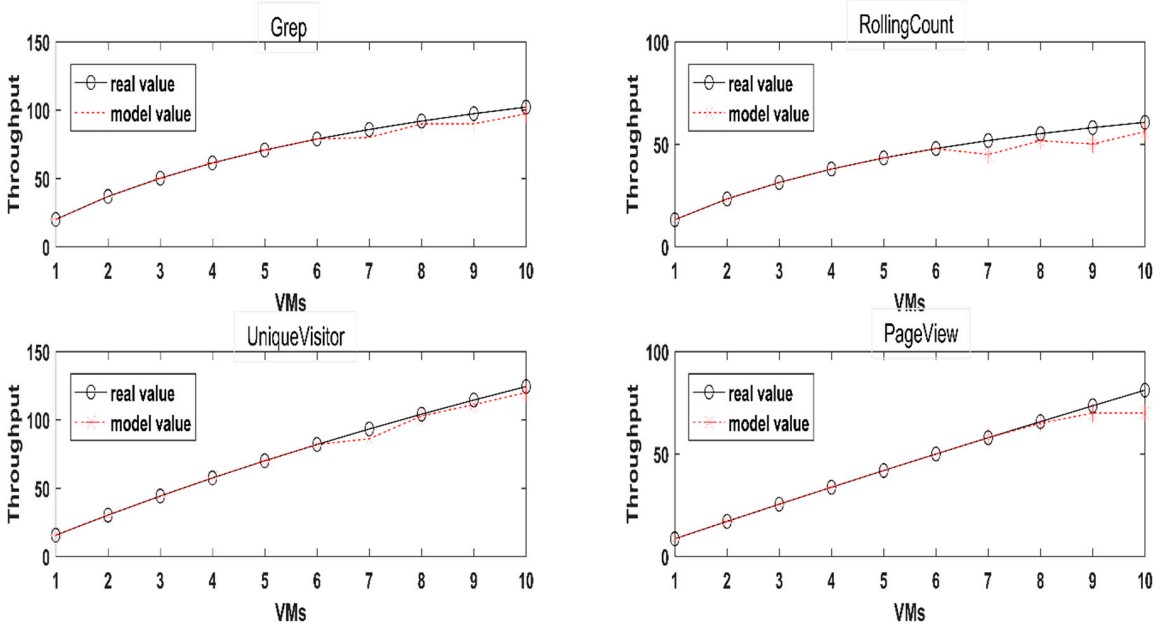

**Figure 2.** MSTW model prediction results.

### 6.4. Results of Elastic Distribution

In order to verify the practicality of the flexible distribution algorithm, we use Producer to simulate the real trace payload (World Cup98) to send Price and Prejudice data to Kafka and select the Grep and Rolling Count two benchmark to handle the data. The time window interval is 5 s, and the prediction algorithm is loaded into the input data. Online prediction, using ganglia to monitor the response time of the application, is used. Due to the prediction error of the load forecasting algorithm and the MSTW model, the appropriate multi configuration virtual machine will also be used in the experiment. In this paper, SLA default is defined as $F > MSTW_{SLA}(m, T)$. Figure 3 shows the application results of the flexible resource distribution algorithm when Grep and Rolling Count process data. Small amount of excessive distribution can be observed at both applications, but the SLA default is 0 and 1 respectively. That is to say, the flexible distribution algorithm can guarantee low SLA default or even no SLA default, so the algorithm can improve the utilization of cloud resources and guarantee the stable processing performance of the system.

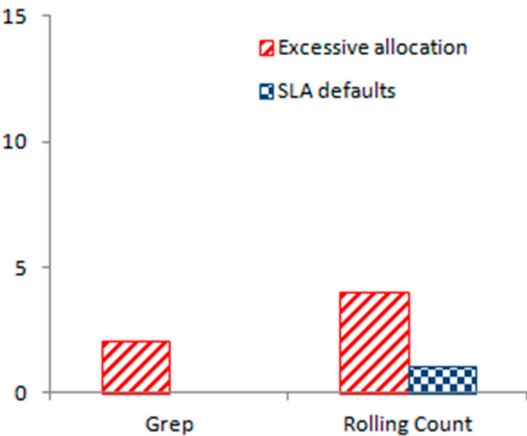

**Figure 3.** Elastic distribution results.

## 7. Discussion

Elastic resource distribution is a hotspot in cloud data center technology research. In the beginning, this paper improved online SVR to predict the load time sequence of stream data and predicted the future time window data load based on historical data load, so that resources can be allocated to reduce cloud resource distribution delay in advance. Then, using to the relationship between the number of virtual machines in the time window and the maximum throughput in the time window, it finally gave a flexible distribution algorithm according to the stream data load prediction value and MSTW performance model to guarantee the SLA of the system.

The key point of this study is to guarantee the system SLA but fails to consider the subsequent impact of the virtual machine distribution or recovery; If the workload suddenly changes (a traffic surge), it can't allocate cloud resources as effectively as usual. In the next step, the author will consider the problem of increasing or reducing the energy consumption of the cloud data center when the virtual machine is increased or reduced, and it also will use another new algorithm to deal with the major variables.

## 8. Patents

Name of patent: A elastic resource allocation method and system for streaming data load. Number:201811381329.1.

**Author Contributions:** H.K.; methodology, software, validation, formal analysis, investigation, data curation, Writing—Original Draft preparation. Z.H. and M.Z.; project administration, supervision, funding acquisition. H.K. and M.Z.; resources, Writing—Review and Editing, visualization

**Funding:** This research was funded by National Natural Science Foundation of China (No. 61602525 and 61572525).

**Acknowledgments:** The authors are grateful to the virtual machine experimental platform of school of software, central south university for their help.

**Conflicts of Interest:** The authors declare no conflict of interest.

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
