# Peer review of "Stream Data Load Prediction for Resource Scaling Using Online Support Vector Regression"

_algorithms, doi:10.3390/a12020037_

Round 1

Reviewer 1 Report

This study proposes online support vector regression for stream data load prediction. The paper presents the time window resource elasticity scaling algorithm and demonstrates its performance based on the numerical experiments. In general, I think that the paper fits well into the scope of the journal. However, some revisions are required before the paper can be considered for publication. Certain segments of the paper must be strengthened. Also, please proofread the paper for English. Below please find more specific comments:

Page 1 line 11: Please define abbreviation “QoS”. Also, there some other abbreviations that are not defined in the abstract (e.g., “”SVR, “SLA”). Please avoid using many abbreviations in the abstract.

Page 1 line 16: I believe “TWRESF” should be replaced with “TWRES”.

*Pages 1-2: Although the paper discusses the relevant literature in the introduction section, I suggest creating a separate section, where the literature will be discussed in details and the gaps in the state-of-the-art should be outlined. Also, how these gaps are addressed in this study should be clearly stated as well.

*The literature review must be strengthened. Towards the end of the section devoted to the literature, I suggest adding a brief discussion regarding importance of advanced solution algorithms for solving challenging decision problems and provide some references to recent publications from reputable Operations Research and Artificial Intelligence journals, including the following:

Canales-Bustos, L., Santibañez-González, E., & Candia-Véjar, A. (2017). A multi-objective optimization model for the design of an effective decarbonized supply chain in mining. International Journal of Production Economics, 193, 449-464.

Dulebenets, M. A. (2018). A Comprehensive Evaluation of Weak and Strong Mutation Mechanisms in Evolutionary Algorithms for Truck Scheduling at Cross-Docking Terminals. IEEE Access, 6, 65635-65650.

Peres, I. T., Repolho, H. M., Martinelli, R., & Monteiro, N. J. (2017). Optimization in inventory-routing problem with planned transshipment: A case study in the retail industry. International Journal of Production Economics, 193, 748-756.

Dulebenets, M. A., Golias, M. M., & Mishra, S. (2018). A collaborative agreement for berth allocation under excessive demand. Engineering Applications of Artificial Intelligence, 69, 76-92.

Fonseca, G. B., Nogueira, T. H., & Ravetti, M. G. (2018). A hybrid Lagrangian metaheuristic for the cross-docking flow shop scheduling problem. European Journal of Operational Research.

Žulj, I., Kramer, S., & Schneider, M. (2018). A hybrid of adaptive large neighborhood search and tabu search for the order-batching problem. European Journal of Operational Research, 264(2), 653-664.

Dulebenets, M. A. (2018). A comprehensive multi-objective optimization model for the vessel scheduling problem in liner shipping. International Journal of Production Economics, 196, 293-318.

*Page 3 line 91: “stream data load;” should be replaced with “stream data load.” The same comment applies to other sentences in this paragraph.

*Page 4 line 128: Why the presented set of equations is not numbered? Please number all the equations in the manuscript.

*Page 11: The discussions section should be strengthened. The authors should clearly highlight limitations of this study and how they will be addressed in future research.

Author Response

Point 1: Page 1 line 11: Please define abbreviation “QoS”. Also, there some other abbreviations that are not defined in the abstract (e.g., “” SVR, “SLA”). Please avoid using many abbreviations in the abstract.

Response 1: Distributed data stream processing system handles real-time, changeable and sudden streaming data load. Its elastic resource allocation has become a fundamental and challenging problem with a fixed strategy will result in waste of resource or a reduction in QoS (quality of service). Spark Streaming as an emerging system is developed to process real time stream data analytics by using micro-batch approach. In this paper, first, we propose an improved SVR (support vector regression) based stream data load prediction scheme. Then, we design a spark-based maximum sustainable throughput of time window (MSTW) performance model to find the optimized number of virtual machines. Finally, we present a resource scaling algorithm TWRES (time window resource elasticity scaling algorithm) with MSTW constraint and streaming data load prediction. The evaluation results show that TWRES could improve the resource utilization effectively and mitigate SLA (service level agreement) violation.

Point 2: Page 1 line 16: I believe “TWRESF” should be replaced with “TWRES”.

Response 2: TERESF has been replaced with TWRES.

Point 3: Pages 1-2: Although the paper discusses the relevant literature in the introduction section, I suggest creating a separate section, where the literature will be discussed in details and the gaps in the state-of-the-art should be outlined. Also, how these gaps are addressed in this study should be clearly stated as well.

Response 3: Cloud computing can add on-demand elasticity to these stream processing systems to deal with fluctuating computing demand using autoscaling[1]. To guarantee a service level agreement (SLA) in terms of application performance, we need a prediction model that connects VM configurations and application performance so that the auto-scalers can estimate and provision the right number of VMs. (Imai, Stacy, and Carlos A. 2017)[2] proposed a maximum sustainable throughput (MST) prediction models for stream processing systems. The models assume a homogeneous VM instance type and predict MST only by the number of VMs, but it not considers the time window interval performance. The paper strengthens the MST to the maximum sustainable throughput of time window (MSTW). If the incoming data rate exceeds the system’s MSTW, unprocessed data accumulates, eventually making the system inoperable. By dynamically allocating and deallocating VMs, service providers can keep the MSTW larger than the incoming data load, to maintain stable service operation. Techniques that have been used to predict data load time series are autoregressive moving average model ARMA [3], neural network [4], and support vector regression SVR [5]. However, these methods are not real online predictions and cannot handle real-time stream data.Our solution responds to the problem by applying a more accurate online SVR algorithm to predict the incoming data-flow load and then measuring the number of virtual machines for the maximum sustainable throughput of time window (MSTW).

Point 4: The literature review must be strengthened. Towards the end of the section devoted to the literature, I suggest adding a brief discussion regarding importance of advanced solution algorithms for solving challenging decision problems and provide some references to recent publications from reputable Operations Research and Artificial Intelligence journals, including the following

Response 4: Recently, some new algorithms for solving different problems of various fields have appeared in the artificial intelligence and reputable Operations Research field. (Canales-Bustos et al.2017)[22] suggested a Pareto-based algorithm, which is a Multi-objective Hybrid Particle Swarm Optimization metaheuristic for the design of an effective decarbonized supply chain in mining. A novel heuristic algorithm, which accounts for the truck service priority and the truck service order restrictions, was proposed for initializing the chromosomes and population[23]. (Peres et al.2017)[24]presented an exact formulation and developed a metaheuristic to solve a multi-period, multi-product Inventory-Routing Problem. A Memetic Algorithm was developed to minimize the total vessel service cost for the vessel diversion decision making[25]. (Dulebenets et al.2018)[26]proposed a hybrid method based on a Lagrangian relaxation technique through the volume algorithm. Using information from the Lagrangian multipliers, constructive heuristics with local search procedures to reach the objective of minimizing the makespan in cross-docking. (Fonseca et al 2018)[27] developed a metaheuristic hybrid based on adaptive large neighborhood search and tabu search, called ALNS/TS, which assume a higher number of customer orders and higher capacities of the picking device. (Žulj et al.2018)[28] proposed a multi-objective mixed integer nonlinear optimization model for the vessel scheduling problem.

A few Evolutionary Algorithms had been developed to solve the stream processing and resource allocation problem. Aiming at the characteristics of real-time streaming data and higher value of new data, this paper proposes a new online support vector regression algorithm, which gives different values to different time periods and improves the accuracy of data load prediction.

Point 5: *Page 3 line 91: “stream data load;” should be replaced with “stream data load.” The same comment applies to other sentences in this paragraph.

Response 5: I’m sorry that i can’t understand this point. If the reviewer gives me more details, I can do better.

Point 6: *Page 4 line 128: Why the presented set of equations is not numbered? Please number all the equations in the manuscript.

Response 6: The equations unnumbered is a constraint condition.

Point 7: *Page 11: The discussions section should be strengthened. The authors should clearly highlight limitations of this study and how they will be addressed in future research.

Response 7: The key point of this study is to guarantee the system SLA but fails to consider the subsequent impact of the virtual machine distribution or recovery; If the workload suddenly changes (a traffic surge), it cant allocate cloud resource gracefully as usual. In the next step, the author will consider the problem of increasing or reducing the energy consumption of the cloud data center when the virtual machine is increased or reduced, and it also will use another new algorithm to deal with the highly variable.

Reviewer 2 Report

The article presents new contents. 

An extensive english editing is necessary.

The article is written in a very chaotic way.

The introduction must introduce the reader into the problem. The authors list several related works, but it is not clear what are the limits of the algorithms in literature and how the proposed algorithm overcomes them. The goal of the article must be clear in the introduction. Acronyms must be defined before using them. 

SVR must be presented before its extension is described. 

Author Response

Point 1: An extensive English editing is necessary.

Response 1: Some grammar errors and tenses have been corrected.

Point 2: The article is written in a very chaotic way.

Response 2: I have created a separate section about introduction and related work.

Point 3: The introduction must introduce the reader into the problem. The authors list several related works, but it is not clear what are the limits of the algorithms in literature and how the proposed algorithm overcomes them. The goal of the article must be clear in the introduction. Acronyms must be defined before using them. 

Response 3: Cloud computing can add on-demand elasticity to these stream processing systems to deal with fluctuating computing demand using autoscaling[1]. To guarantee a service level agreement (SLA) in terms of application performance, we need a prediction model that connects VM configurations and application performance so that the auto-scalers can estimate and provision the right number of VMs. (Imai, Stacy, and Carlos A. 2017)[2] proposed a maximum sustainable throughput (MST) prediction models for stream processing systems. The models assume a homogeneous VM instance type and predict MST only by the number of VMs, but it not considers the time window interval performance. The paper strengthens the MST to the maximum sustainable throughput of time window (MSTW). If the incoming data rate exceeds the system’s MSTW, unprocessed data accumulates, eventually making the system inoperable. By dynamically allocating and deallocating VMs, service providers can keep the MSTW larger than the incoming data load, to maintain stable service operation. Techniques that have been used to predict data load time series are autoregressive moving average model ARMA [3], neural network [4], and support vector regression SVR [5]. However, these methods are not real online predictions and cannot handle real-time stream data.Our solution responds to the problem by applying a more accurate online SVR algorithm to predict the incoming data-flow load and then measuring the number of virtual machines for the maximum sustainable throughput of time window (MSTW). 

Distributed data stream processing system handles real-time, changeable and sudden streaming data load. Its elastic resource allocation has become a fundamental and challenging problem with a fixed strategy will result in waste of resource or a reduction in QoS (quality of service). Spark Streaming as an emerging system is developed to process real time stream data analytics by using micro-batch approach. In this paper, first, we propose an improved SVR (support vector regression) based stream data load prediction scheme. Then, we design a spark-based maximum sustainable throughput of time window (MSTW) performance model to find the optimized number of virtual machines. Finally, we present a resource scaling algorithm TWRES (time window resource elasticity scaling algorithm) with MSTW constraint and streaming data load prediction. The evaluation results show that TWRES could improve the resource utilization effectively and mitigate SLA (service level agreement) violation.

Point 4: SVR must be presented before its extension is described.

Response 4: SVR is presented in the literature.

Round 2

Reviewer 1 Report

The authors adequately addressed my comments and I recommend this study for publication.    

Reviewer 2 Report

The authors have modified the article according to the reviewer's suggestions.